# Crop Yield Prediction Using Hybrid Machine Learning Approach: A Case Study of Lentil (*Lens culinaris* Medik.)

Pankaj Das [1], Girish Kumar Jha [2],*, Achal Lama [1] and Rajender Parsad [1]

1 ICAR-Indian Agricultural Statistics Research Institute, New Delhi 110012, India
2 ICAR-Indian Agricultural Research Institute, New Delhi 110012, India
* Correspondence: girish.stat@gmail.com

**Abstract:** This paper introduces a novel hybrid approach, combining machine learning algorithms with feature selection, for efficient modelling and forecasting of complex phenomenon governed by multifactorial and nonlinear behaviours, such as crop yield. We have attempted to harness the benefits of the soft computing algorithm multivariate adaptive regression spline (MARS) for feature selection coupled with support vector regression (SVR) and artificial neural network (ANN) for efficiently mapping the relationship between the predictors and predictand variables using the MARS-ANN and MARS-SVR hybrid frameworks. The performances of the algorithms are com-pared on different fit statistics such as RMSE, MAD, MAPE, etc., using numeric agronomic traits of 518 lentil genotypes to predict grain yield. The proposed MARS-based hybrid models outperformed individual models such as MARS, SVR and ANN. This is largely due to the enhanced feature ex-traction capability of the MARS model coupled with the nonlinear adaptive learning ability of ANN and SVR. The superiority of the proposed hybrid models MARS-ANN and MARS-SVM in terms of model building and generalisation ability was demonstrated.

**Keywords:** soft computing; MARS; SVM; ANN; hybrid approach

## 1. Introduction

Globally, pulses are the second most important crop group after cereals. Lentil is one of the most widely consumed pulses in India and specifically in the Middle East and South Asian regions [1]. Being a rich source of essential nutrients, it is regarded as a high value crop for ensuring food and nutritional security for millions of people in developing countries. It is drought-tolerant and can also be grown as a rotation crop. Lentils also improve soil fertility by replenishing soil nitrogen levels. [2]. India contributes around 18% to world lentil production and is one of the major lentil-exporting countries in the world [3].

Despite being a major producer and consumer, the yield of lentil is considerably low in India compared to other major producing countries. The crop yield is affected by multiple factors such as physical, economic and technological. Morphological characters play a crucial role in yield enhancement as well as reduction. [4,5]. A good prediction model explores the complex relationship between different factors and yield. It helps to improve management techniques and boost actual yields. A good prediction model should be reliable, consistent, object-oriented, cost effective and sensitive to extreme events [6]. Several researchers have attempted to model the crop yield of lentil using different models such as simple correlation [1], path analysis [7], multiple linear regression [8], stepwise regression [9], factorial analysis [2] and principle component analysis [10]. These studies assumed the linear relationship between plant characters and crop yield. However, these models have not been successful in capturing the nonlinear relationship between crop yield and plant characters [11].

In the past decades, there has been a consistently rising interest in the application of machine learning (ML) techniques such as artificial neural networks (ANNs), support

vector regression (SVR) and random forest (RF) in different fields, particularly for modelling nonlinear relationships. Schultz and Wieland [12] discussed the possibilities of applying neural networks or neural networks in combination with fuzzy techniques in the field of agroecological modelling. Uno et al. [13] used artificial neural networks to predict corn yield from compact airborne spectrographic imager data. They used statistical and ANN approaches along with various vegetation indices to develop yield prediction models. Lee et al. [14] and Zhang et al. [15] found that multivariate adaptive regression spline (MARS) performed better than both statistical parametric methods such as linear discriminant analysis or logistic regression and nonparametric approaches such as neural networks and support vector machines. Khazaei et al. [16] applied artificial neural network methodology to model the correlation between crop yield and 10 yield components of chickpea (*Cicer arietinum* L.). They also used the fuzzy c-means clustering technique for the classification of 362 chickpea genotypes based on agronomic and morphological traits. Among the various ANN structures, the 10-14-3-1 ANN structure with a training algorithm of back-propagation and hyperbolic tangent transfer function in the hidden and output layers performed best. Higgins et al. [17] developed an ANN model for forecasting the maturity of green peas using historical harvest information along with weather and climate forecasts. They implemented and evaluated the model in a large pea growing region in Tasmania, Australia. The model allowed for not only the harvesting of peas closer to their ideal maturity indices, but also the planning of harvest and transportation logistics with a significantly longer lead time. Khairunniza-Bejo et al. [11] highlighted the prediction accuracy of the ANN model compared to other linear models in crop yield prediction. They showed that the ANN model captured the relationship among the variables much more accurately. Gandhi et al. [18] used the support vector machine (SVM) model for rice crop yield prediction in India using climatic variables. Deo et al. [19] applied MARS, least square, SVM and decision tree for drought forecasting in eastern Australia. Garg et al. [20] forecast rice yield using the fuzzy logic and regression model. They tested four different types of the fuzzy interval with four degrees of regression equations. Ying-Xue et al. [21] developed a support vector machine-based open crop model (SBOCM) integrating developmental stage and yield prediction models for rice crop yield prediction in China. Klompenburg et al. [22] and Batool et al. [23], Cubillas et al. [24], Bali and Singla [25] and Ji et al. [26] reviewed the research works related to crop yield prediction using ML techniques. They explored the different machine learning techniques used in crop yield prediction and their efficiency. The studies reported the increasing trend of hybrid models in crop yield prediction.

The selection of appropriate input variables is an important part of any model such as multiple linear regression models (MLRs) and machine learning models [27–29]. Feature selection is an effective way to reduce computation time, improve learning accuracy and facilitate a better understanding of the learning model or data. Many studies [30–32] suggested that variable selection reduces the complexity of the model and make it more interpretable. However, each variable (feature) selection strategy is data-based and has its own benefit, drawback and applicability. MARS is based on local regression modelling, which uses spline functions to approximate complex nonlinear relations [33,34]. The advantage of MARS is that the relative importance of independent variables to the dependent variable can be captured [15,33,35]. The important variable can be selected based on the relative importance of independent variables. Therefore, MARS was used as a selection model in the present study.

In the literature, most researchers have restricted themselves to using only one method such as ANN in their study. Comparative study and hybrid modelling of soft computing techniques with variable selection on particular datasets is yet to be done. This motivated the present comparative study of different soft computing techniques such as ANN, MARS and SVR. A hybrid model was formulated using MARS and ANN/SVR. These techniques and the proposed hybrid model were applied to the lentil dataset, and their modelling and forecasting performances were compared using different statistical measures. The remain-

ing portion of the paper is divided into materials and methods, results and discussion, and a conclusion section.

## 2. Materials and Methods

### 2.1. Plant Material and Field Experiment

The experimental data for this study comprise 518 lentil accessions, of which 206 entries are exotic collections and 312 are indigenous collections, including 59 breeding lines. These accessions were grown in augmented block design with five checks during rabi season, 2006–07 at ICAR-Indian Institute of Pulses Research, Kanpur. Accessions were evaluated for 21 descriptors, including plant characteristics and seed characteristics following the biodiversity and national Distinctness, Uniformity and Stability (DUS) descriptors guidelines. More information on the descriptors is accessible in [36]. Ten numerical descriptors including days to 50% flowering (DF), plant height (PH), days to 90% maturity (DM), 100 seed weight (SW), biological yield per plant (BYP), number of primary branch per plant (PB), number of secondary branch per plant (SB), number of pods per plant (PPP), yield per plant (YPP) and plant height at lowest pod (PHLP) were used for detailed analysis. Our target variable was yield per plant (YPP), which was influenced by other factors (See Supplementary Materials).

### 2.2. Machine Learning Models

#### 2.2.1. Multivariate Adaptive Regression Spline (MARS) Model

The MARS model for a dependent (outcome) variable y, and M terms, can be summarized in the following equation [37]:

$$\hat{\vec{y}} = \hat{f}_M(\vec{X}) = c_0 + \sum_{m=1}^{M} c_m B_m(\vec{X}) \tag{1}$$

where $\hat{\vec{y}}$ is the dependent variable (YPP) predicted by the MARS model, $c_0$ is a constant, $B_m(\vec{X})$ is the $m$th basis function, which may be a single spline basis function, and $c_m$ is the coefficient of the $m$th basis function. Both the variables to be introduced into the model and the knot positions for each individual variable have to be optimized. For a dataset $\vec{X}$ containing $n$ objects and $p$ explanatory variables, there are $N = n \times p$ pairs of spline basis functions with knot locations ($i = 1, 2, \ldots, n; j = 1, 2, \ldots, p$). In the present study, $x = 206$ entries are exotic collections and 312 are indigenous collections, including 59 breeding lines, $p = 9$ and $n = 518$. MARS was used as a variable selection model in the present study.

#### 2.2.2. Artificial Neural Network (ANN) Model

Artificial neural networks (ANNs) are nonlinear data-driven self-adaptive approaches as opposed to the traditional model-based methods [38]. ANNs have the ability to identify correlations between input variables and associated target values. ANNs can solve problems involving nonlinear and complicated data even if the data are noisy and inaccurate since they mimic the learning process of the human brain. They are therefore perfectly suited for modelling agricultural data, which are known to be complicated and frequently nonlinear.

The output of a neural network can be expressed by the following equation [39]:

$$y_t = \alpha_0 + \sum_{j=1}^{n} \alpha_j f(\sum_{i=1}^{m} \beta_{ij} y_{t-1} + \beta_{oj}) + \varepsilon_t \tag{2}$$

where $y_t$ is output of neural network model (yield per plant); $n$ is number of hidden nodes; $m$ is the number of input nodes; f is the net input of the activation function; $\beta_{ij}$ {$I = 1, 2, \ldots, m$; $j = 0, 1, \ldots, n$} are the weights from input to hidden nodes; $\alpha_j$ {$j = 0, 1, \ldots, n$} are the vectors of weights from hidden to output node; $\alpha_0$ and $\beta_{0j}$ are the weights of arcs leading from bias terms. Activation function is a differentiable function that is used for smoothing the

result of the cross product of the covariate or neurons and the weights. In artificial neural networks, the activation function of a node defines the output of that node given an input or set of inputs. In the present study, logistic function was used as activation function and the Levenberg–Marquardt (LM) learning algorithm was used to adjust the weights in the multi-layered feedforward networks.

### 2.2.3. Support Vector Regression (SVR) Model

Support Vector Machine (SVM) is nonlinear algorithms used in supervised learning frameworks for data analysis and pattern recognition [8]. The traditional support vector machine technique is the predecessor of the SVR that is a nonlinear prediction model. Based on Vapnik's concept of support vectors [8], Drucker et al. [40] first introduced support vector regression. SVR's primary objective is to maximize the difference between predicted and actual values while minimizing error by adding a hyperplane. The SVR model can be written as:

$$y = \sum_{i=1}^{N} W_i Ker(x_i, x_j) + b \tag{3}$$

where $y$ is our dependent variable (YPP), $W_i$ is associated weight and $Ker(x_i, x_j)$ is the nonlinear mapping function known as kernel function for input (independent) variables $x_i$ ($I = 1, 2, \ldots, 9$).

### 2.2.4. Hyperparameter Tuning of Machine Learning Models

Hyperparameter is one of the important factors in the ML model's accuracy and prediction. Hyperparameters work differently in different datasets [17,41]. Choosing the best hyperparameter values improves the stability of any ML performance. Randomly, a ratio of 80 and 20 of the whole dataset was used for the training sample and the testing sample, respectively. The k fold cross-validation method was adopted to deal with the model over fitting problem. The cross-validation method evaluates ML algorithms' capacity to handle new and unexplored data. In this method, the dataset is divided into approximately k equal-sized groups (folds) at random. The data are trained on k-1 folds, with the first fold being treated as a testing set. For each dataset in this investigation, three folds (k = 3) were taken. Some studies [13,21,42,43] suggest the superiority of this approach for a small dataset. The details of used hyperparameters for the used ML models are provided in Table 1. The hyperparameter tuning was conducted based on the trial-and-error method for minimizing root mean square error (RMSE).

**Table 1.** Hyper parameter tuning of the machine learning model.

| Model | Hyper Parameters | |
|---|---|---|
| ANN | Training algorithm | Resilient back propagation (Rprop) |
| | Maximum steps up to which the neural network is trained (*Stepmax*) | $1 \times 10^7$ |
| | The number of repetitions used to train the neural network model (*Rep*) | 3 |
| | Threshold (threshold value of the partial derivatives of the error function) | 0.01 |
| SVR | Defining algorithms | Kernel |
| | Regularization parameter (*C*) | 1 |
| | Kernel coefficient (*Gamma*) | Scale |
| | Penalty function (*Epsilon*) | 0.1 |
| | Cross validation | 10 |
| MARS | Number of model terms upper bound (*Nmax*) | 20 |
| | Penalty coefficient (*b*) | 3 |

### 2.3. Proposed Mars Based Hybrid Model

In the present study, MARS-based hybrid models have been developed by combing them with ANN and SVR, respectively. The main motive to develop these hybrid models was to harness the variable selection ability of MARS algorithm and prediction ability of ANN/SVR simultaneously. First, MARS algorithm was used to find important variables among the independent variables that influences yield variable. Then these selected variables were taken as input variables to predict yield variable (Figure 1). The pictorial representation of the hybrid methodology can be summed up in the following steps:

Step 1. Start model building with all available predictors.
Step 2. Apply MARS algorithm for extracting the important predictors based on its importance.
Step 3. Build the machine learning model (ANN/SVR) using the selected predictors.
Step 4. Obtain prediction using the model obtained in Step 3.

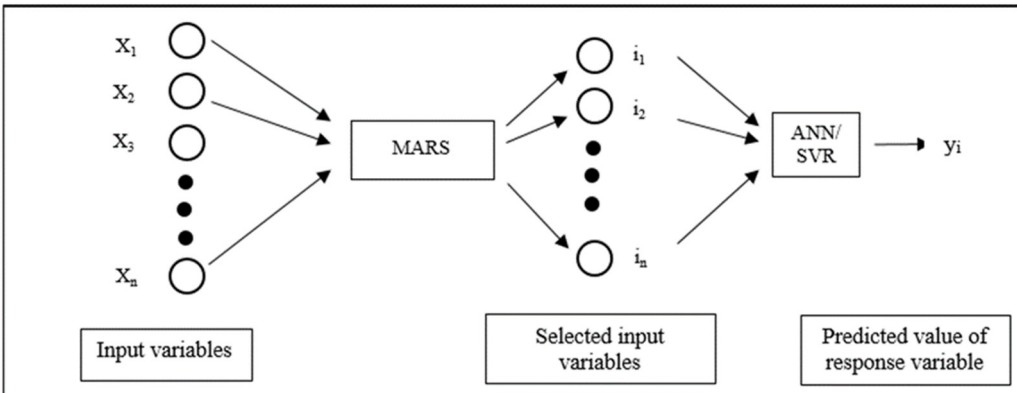

**Figure 1.** MARS-based ANN/SVR hybrid framework.

### 2.4. Model Performance and Accuracy of Fitted Models

The model accuracy measures for root mean squared error (RMSE), mean absolute deviation (MAD), mean absolute percentage error (MAPE) and maximum error (ME) were used to select the best models. The formulas were used as follows:

$$RMSE = \sqrt{\frac{\sum_{i=1}^{N}(y_i - \hat{y}_i)^2}{N}} \qquad MAD = \frac{\sum_{i=1}^{N}|y_i - \hat{y}_i|}{N}$$

$$MAPE = \frac{\sum_{i=1}^{N}|y_i - \hat{y}_i|/y_i}{N} \qquad ME = \max \sum_{i=1}^{N}|y_i - \hat{y}_i| \tag{4}$$

where $y_i$ and $\hat{y}_i$ are the actual value and predicted value of response variable and $N$ is the number of data. The Diebold Mariano (DM) test [44] was used in addition to accuracy measurements to compare the final fitted models.

### 3. Results

In this study the MARS, ANN and SVR model was fitted with the help of R. Two new R packages i.e., "MARSANNhybrid" [45] and "MARSSVRhybrid" [46] were developed for fitting of the MARS-based ANN and SVR models, respectively. The hyperparameters of the fitted models are described in Table 1.

### 3.1. Data Processing and Statistical Analysis

The basic aim of model building is to find out the existence of a relationship between the output and input variables. The summary statistics such as mean, range, standard deviation and coefficient of variation (CV) of parameters were checked (Table 2). The lowest

coefficient of variation was observed in DM character, while dependent YPP showed the highest variation. Among the independent variables, the highest variability was found in the number of pods per plant (PPP).

**Table 2.** Summary of the parameters of lentil.

| Parameter | Range | Mean | Standard Deviation | CV |
|---|---|---|---|---|
| DF | 58–106 | 78.69 | 10.75 | 13.66 |
| PH | 17–47.6 | 30.79 | 4.79 | 15.55 |
| DM | 114–140 | 126.03 | 4.70 | 3.72 |
| SW | 1.2–4.1 | 2.43 | 0.53 | 21.75 |
| BYP | 4.2–28 | 13.37 | 3.75 | 28.01 |
| PB | 2–9 | 3.76 | 1.06 | 28.20 |
| SB | 4–18 | 10.22 | 2.39 | 23.40 |
| PPP | 3.7–309.3 | 116.16 | 47.53 | 40.92 |
| YPP | 0.2–10.7 | 3.72 | 1.61 | 43.25 |
| PHLP | 1–19 | 10.74 | 2.30 | 21.44 |

The correlation study of input variables with outcome was explored (Figure 1) for the aforementioned lentil dataset. The Pearson correlation coefficients of input variables with output variables helped to identify the plant characters (traits) that have a strong correlation with output. As per Figure 2, there is a significant positive correlation between yields per plant (YPP) and pods per plant (PPP); biological yield per plant (BYP); number of secondary branches per plant (SB); plant height at lowest pod (PHLP); number of primary branches per plant (PB); plant height (PH); and 100 seed weight (SW). In the literature, similar types of correlation results were reported by different researchers who showed that PH [42,47,48], PB and SB [42,47,48], PPP [42], SW [42], PHLP [42] and BYP [42] are closely related to the yield of lentil. We also found a significant negative correlation between YPP and days to 90% maturity (DM) and days to 50% flowering (DF). These negative relationships were confirmed by some researchers [42,49,50].

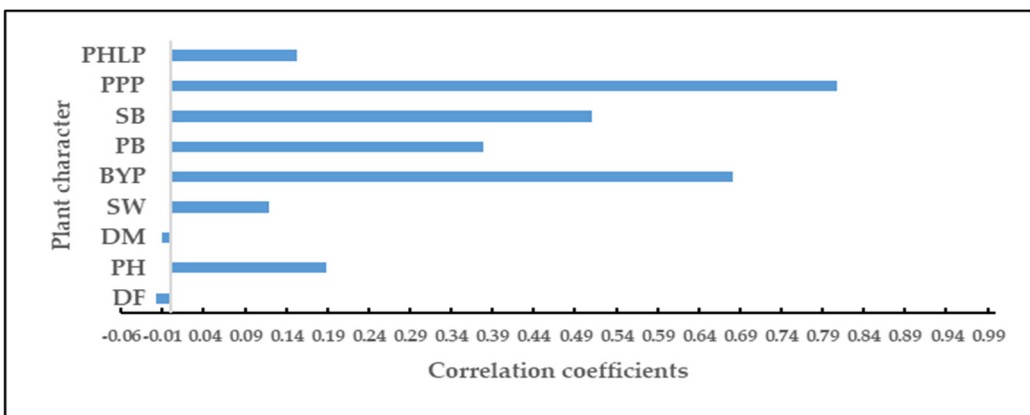

**Figure 2.** Pearson correlation coefficients of input variables with yield per plant of lentil.

Figure 2 depicts the Pearson correlation coefficients of input variables with the yield per plant of lentil. Variables were normalized within ranges [0.1, 1] for fitting in the ML models [51]. The given below formula was used for normalization:

$$x_n = \left( \frac{x_i - x_{\min}}{x_{\max} - x_{\min}} \right) \tag{5}$$

where $x_i$ is the original data, $x_n$ is the normalized values, and $x_{max}$ and $x_{min}$ are the maximum and minimum values, respectively. Denormalised has been done prior to calculation of performance measures.

### 3.2. Input Variable Selection

MARS was used as a variable selection method. MARS degree largely influences the performance of model fitting and forecasting. Hence, we critically examined the performance of the model on different degrees (df 1, 2 and 3). The performance for the MARS model of degree 1, 2 and 3 were evaluated. It was found that the model complexity increased as the MARS degree increased. To compare the model accuracy of these MARS models, RMSE, MAD, MAPE and ME were computed. Table 3 presents the RMSE, MAD, MAPE and ME of the three MARS models. It was found that the MARS model with interaction terms tended to perform better than the model without interaction (MARS model with degree 1). The MARS model with degree of 3 had smaller RMSE, MAD and MAPE values compared to the MARS model with degree of 1 and 2. Moreover, the performance of the MARS model with degree of 3 followed all the assumptions of model adequacy, while the other two models failed to satisfy those assumptions. Therefore, the MARS model with degree of 3 was selected for the model fitting. The selected equation of the MARS model is:

$$Y(YPP) = 1.85 - 8.43 \times BF1 - 0.15 \times BF2 + 0.03 \times BF3 + 0.29 \times BF4 + 4.06 \times BF5 + 0.04 \times BF6 \\ -0.01 \times BF7 + 0.01 \times BF8 - 0.001 \times BF9 + 0.002 \times BF10 - 0.21 \times BF11 - 0.0002 \times BF12 \tag{6}$$

**Table 3.** Performance measures for different MARS models.

| Degree | RMSE | MAD | MAPE |
|:---:|:---:|:---:|:---:|
| 1 | 0.5972 | 0.5134 | 0.1792 |
| 2 | 0.4492 | 0.4866 | 0.1566 |
| 3 | 0.4356 | 0.4842 | 0.1565 |

On the basis of generalized cross-validation (GCV) and residual sum of squares (RSS), a MARS model of order 3 was built to extract the significant variables. In Table 4, the MARS model's chosen variables are listed. The dependent variables were then predicted using the seven variables that were chosen.

**Table 4.** Predictor importance for MARS with degree of 3.

| Variable | GCV | RSS |
|:---:|:---:|:---:|
| PPP | 100 | 100 |
| SW | 46.5 | 48.6 |
| Ph | 31.2 | 33.9 |
| BYP | 31.1 | 33.9 |
| PHLP | 25.5 | 28 |
| PB | 18.4 | 20.1 |
| DF | 6.6 | 7.9 |

### 3.3. ANN Model Development

For model-building purposes, we varied our model architecture with 1 to 5 hidden nodes with a single hidden layer. The resilient backpropagation method was used for model training. In the present study, neural network models were fitted with rep = 1 to 3, stepmax = $1 \times 10^5$ to $1 \times 10^8$ and threshold = 0.01. The schematic representation of the fitted ANN model with weights is shown in Figure 3. Table 5 summarizes the error rate and performance measures of fitted ANN with a different number of hidden nodes. The ANN model with 1, 2, 3 and 5 hidden nodes had the same performance. However, the ANN model with 4 hidden nodes gave the best result. Thus, the best-fitted replication in the ANN model with 4 hidden nodes was used for yield forecasting.

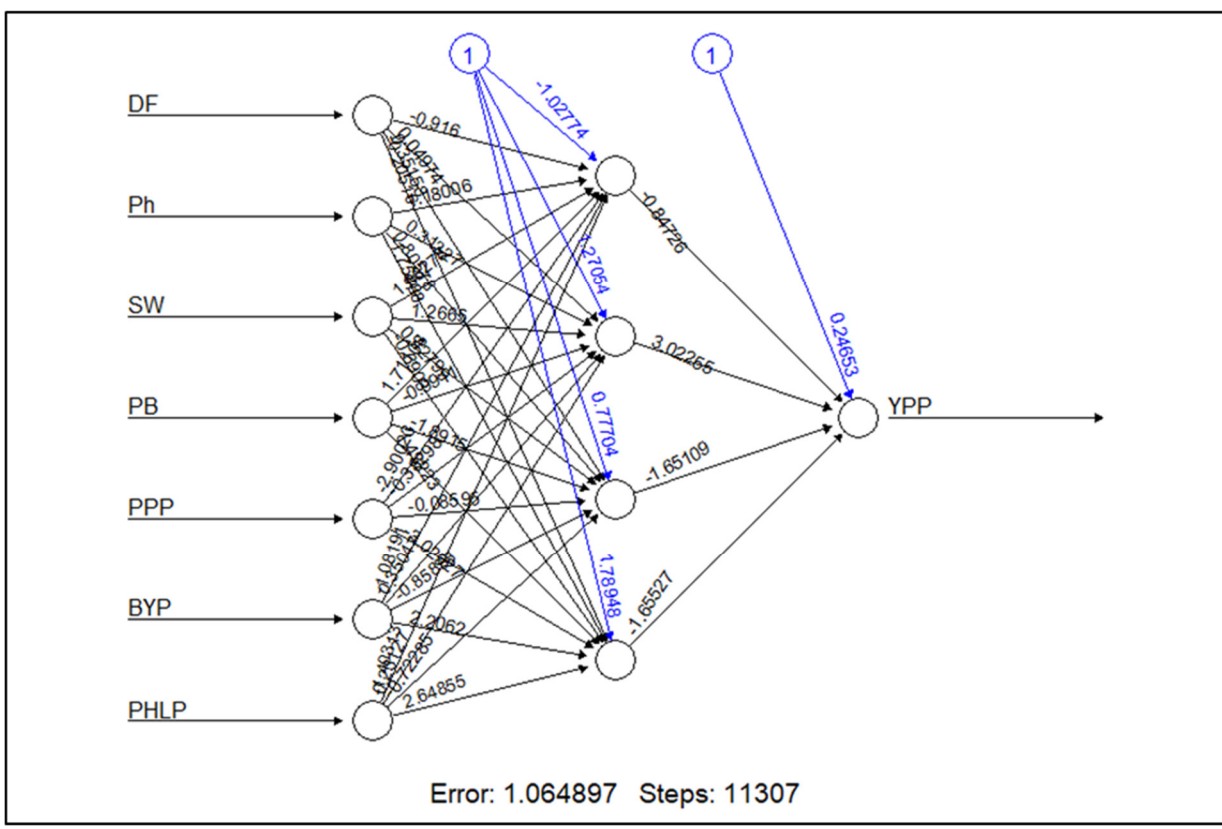

**Figure 3.** Schematic diagram of fitted neural network structure.

**Table 5.** Performance measures for different number of nodes in ANN models.

| No. of Nodes in Hidden Layer | RMSE | MAD | MAPE |
|---|---|---|---|
| 1 | 1.6134 | 1.2169 | 0.4612 |
| 2 | 1.6134 | 1.2169 | 0.4612 |
| 3 | 1.6134 | 1.2169 | 0.4612 |
| 4 | 0.9627 | 0.6288 | 0.1828 |
| 5 | 1.1512 | 1.0191 | 0.3521 |

*3.4. SVR Model Development*

The SVR model was fitted using different types of kernel functions such as linear, radial basis, sigmoid and polynomial, although the most often used and recommended function is radial basis. However, it is recommended to select the appropriate kernel function for the given dataset. Therefore, SVR was fitted using the four different kernel basis functions, and the best model was selected on the basis of performance measures. Other significant hyperparameters in the SVR model, such as the epsilon factor, cross-validation and type of regression, also have a significant impact on the model's performance. In Table 1, the values of these parameters that were used are reported. Tenfold cross-validation was used to validate the resulting model. Further, we focused our attention on selection of the appropriate kernel function. The support vectors produced using RBF, linear and polynomial were 323, 321 and 329, respectively. In the training phase, performance measures such as RMSE, MAD, MAPE and ME were computed for choosing the best forecasting model. From Table 6, it was observed that the SVR model with radial basis kernel function provided the best result on the basis of parsimonious representation. Hence, the SVR model with radial basis kernel function was employed for yield prediction.

**Table 6.** Performance measures for different kernel function in SVM.

| Kernel Function | RMSE | MAD |
|---|---|---|
| Radial basis | 0.6474 | 0.3602 |
| Linear | 0.8599 | 0.5231 |
| Polynomial | 0.827 | 0.5253 |
| Sigmoid | 0.8269 | 0.5253 |

Fit statistics values were used to examine the effectiveness of fitted models for both in-sample and out-of-sample predictions. In order to verify the model's suitability, the specifics of the derived residuals were also examined. The value of the statistic of fitted models is shown in Table 7. It was found that MARS performed the best among the individual models, followed by SVR and ANN. To fit a local regression model to each sub-region, the MARS algorithm partitioned the dataset into a number of sub-regions. Redundancy of variables may cause overfitting problems in a single ANN (without MARS). Similar results in ANN were highlighted by researchers [30–32]. It might be the reason why the ANN performed poorly. The performance of a single SVR model was considerably affected by the same issues.

**Table 7.** Performance measures for different fitted models.

| | Model | RMSE | MAD | MAPE | ME |
|---|---|---|---|---|---|
| | ANN | 0.9827 | 0.6288 | 0.1828 | 8.1055 |
| | MARS | 0.4356 | 0.4842 | 0.1565 | 4.2157 |
| In-sample | MARS-ANN | 0.0802 | 0.0607 | 0.2478 | 0.3918 |
| | MARS-SVR | 0.0826 | 0.0579 | 0.1834 | 0.8498 |
| | MLR | 0.9869 | 0.6520 | 0.1840 | 9.10 |
| | SVR | 0.6474 | 0.3602 | 0.1089 | 7.1265 |
| | ANN | 0.8142 | 0.6435 | 0.2308 | 2.4871 |
| | MARS | 0.9415 | 0.6147 | 0.2769 | 5.3540 |
| Out-sample | MARS-ANN | 0.0802 | 0.0579 | 0.2214 | 0.7085 |
| | MARS-SVR | 0.0658 | 0.0579 | 0.1626 | 0.2206 |
| | MLR | 0.8520 | 0.0610 | 0.2852 | 3.6302 |
| | SVR | 0.6853 | 0.4902 | 0.2707 | 2.6435 |

The out-of-sample performance of these hybrid models further demonstrates their strong generalizability. The DM test was also used to determine whether the MARS-ANN and MARS-SVR models were the best. The alternative MARS-ANN model outperformed the MARS-SVR model in terms of accuracy, which was the null hypothesis of the test. The significance of the Diebold–Mariano (DM) test is displayed in Table 8. The right-tailed DM test was used considering the models have equal accuracy. The test demonstrated the superiority of the MARS-ANN model over the MARS-SVR.

**Table 8.** Results of DM test of compared models.

| Model | DM Value | *p* Value | Remarks |
|---|---|---|---|
| MARS-ANN vs. MARS-SVR | 4.185 | <0.01 | The accuracy of MARS-ANN is better than MARS-SVR. |
| MARS-ANN vs. ANN | 5.304 | <0.01 | The accuracy of MARS-ANN is better than ANN model. |
| MARS-ANN vs. MARS | 5.725 | <0.01 | The accuracy of MARS-ANN is better than MARS model. |

**Table 8.** *Cont.*

| Model | DM Value | *p* Value | Remarks |
|---|---|---|---|
| MARS-ANN vs. SVR | 5.955 | <0.01 | The accuracy of MARS-ANN is better than SVR model. |
| MARS-SVR vs. ANN | 6.563 | <0.01 | The accuracy of MARS-SVR is better than ANN model. |
| MARS-SVR vs. SVR | 6.823 | <0.01 | The accuracy of MARS-SVR is better than SVR model. |
| MARS-SVR vs. MARS | 6.235 | <0.01 | The accuracy of MARS-SVR is better than MARS model. |

## 4. Discussion

The study revealed the superiority of proposed hybrid models for crop yield prediction. The results indicated that the proposed hybrid model had the power to capture the nonlinearity among the variables. The generic models such as ANN, SVR and MARS failed to capture the inherent data patterns and were unable to produce satisfactory prediction results. It is clear that variable selection provided extra advantages to the SVR and ANN models. Both of the proposed hybrid models outperformed their individual counterparts. It validated the advancements made by MARS in both the ANN and SVR models. As previously mentioned, key explanatory variables were retrieved with the aid of the MARS model in the case of hybrid models, and nonlinear forecasting techniques such as ANN and SVR were applied. The superior performance of the hybrid models may be attributable to parsimony and two-stage model construction. Further DM test results clarified MARS-ANN was the best model among the fitted models.

## 5. Conclusions

Modelling and forecasting of complex, multifactorial and nonlinear phenomenon such as crop yield have intrigued researchers for decades. This study is an attempt in the similar direction to contribute to the vast literature of crop-yield modelling. The study proposed novel hybrids based on MARS. The feature extraction ability of MARS was utilized, and efficient forecasting models were developed using ANN and SVR. The utility of the proposed models was illustrated and compared using a lentil dataset with baseline models. As these models do not depend on assumptions about functional form, probability distribution or smoothness and have been proven to be universal approximators.

The novel hybrid model was built in two steps, each performing a specialized task. In the first step, important input variables were identified using the MARS model instead of hand-picking variables based on a theoretical framework. In the second step, nonlinear prediction techniques ANN and SVR were used for yield prediction using the selected variables. The performance of the models was compared using fit statistics such as RMSE, MAD, MAPE and ME. The proposed MARS-based hybrid models performed better as compared to the individual models such as MARS, SVR and ANN. This is largely due to the enhanced feature extraction capability of the MARS model coupled with the nonlinear adaptive learning feature of ANN and SVR. This proposed framework can be applied to a variety of datasets to capture the nonlinear relationship between independent and dependent variables. The R packages developed in this study have utility in multifactorial and multivariate experiments such as genomic selection, gene expression analysis, survival analysis, digital soil mappings, etc. Further, efforts can be directed to propose and evaluate hybrids of other soft computing techniques.

**Supplementary Materials:** The following are available online at https://www.mdpi.com/article/10.3390/agriculture13030596/s1. Supplementary sheet for Descriptors.

**Author Contributions:** P.D. conceived the conceptualization, investigation, formal analysis, data curation and writing original draft. G.K.J. gave the idea of conceptualization, resources, reviewing and editing. A.L. and R.P. performed supervision and edited the manuscript. All authors have read and agreed to the published version of the manuscript.

**Funding:** This research was funded by ICAR-Indian Agricultural Statistics Research Institute, New Delhi, India.

**Institutional Review Board Statement:** Not applicable.

**Data Availability Statement:** The data presented in this study are available on request from the corresponding author.

**Acknowledgments:** The authors are thankful to the Director, ICAR-IASRI for providing facilities for carrying out the present research.

**Conflicts of Interest:** The authors declare no conflict of interest.

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
