# Peer review of "Crop Yield Prediction Using Hybrid Machine Learning Approach: A Case Study of Lentil (Lens culinaris Medik.)"

_agriculture, doi:10.3390/agriculture13030596_

Round 1

Reviewer 1 Report

This work tell me a story about the comparison of the model fitness and prediction effect of three prediction models and two hybrid prediction models, and found that hybrid models MARS-ANN and MARS-SVM have superiority over single model. However, I can’t judge the novelty of this work because the author didn’t provide whether such work has been conducted in previous publications, and what’s the new improvement compared with previous findings. Moreover, the article is badly written and is not quite formal. Therefore, I recommend the author supplement some important contents in almost every part of the manuscript, and resubmit for further evaluation.

The detailed questions and comments that need improvement are as follows:

1, line 8-22 in the abstract part are the background, and should be shorted to two to three lines, show more contents on your main results and new findings. In the abstract, the two R packages are not developed by this work, and should not be mentioned in abstracts. Please show how MARS-ANN and MARS-SVM have better performance.

2, In the introduction, in line 33, the word “share” seems improper; pay attention to the first letter of “Morphological” and “it” in line 36, such grammatical errors are prevalent in the manuscript. Line 30-46 is the first paragraph (para), the author tell us too many things, but not focal, I would suggest the author divided it into to part, with the first describing the importance of lentil, and the second about the linear models along with their shortcomings. Line 47-82 is the second para, the author raised many examples, but didn’t introduce whether the methods used in this study have been used before, and what should followers do to make improvements. Therefore, I can’t see the necessity and meaning of this work. Moreover, the author should introduce these models in this para. Line 83-91 should tell the objective of this study, especially their new ideas or thinkings. Bear in mind that the full names of each abbreviation should be provided when it first appears.

3, in line 94-96, if this data have been used in other publications, it is better to cite the references;  in line 98, if the author use all the 21 descriptors, the author should provide these descriptors, and how they were measured as supplementary (supp). Some part (Line 108-120, 126) of “2.2. Data processing, statistical analysis and input variable selection” should be moved to the part of ”results”, if these results were not provided in other findings (if yes, these results should be removed or briefly introduced). Some part (Line 120-125) should be moved to “discussion”. Some part (Line 134-145) should be moved to “introduction”.  Some part (Line 153-155) should be moved to the first part of “Materials of Methods”.

4, what is the relationship between “2.3.1. Multiple linear Regression (MLR) model” and this work, many manuscripts can copy and use lines 158-162.  Lines 164-170 should be move to “Introduction”.  The descriptions of “n is number of hidden nodes, m is the number of hidden nodes” in lines 172-173 indicate that this is a careless manuscript.

Please check the whole manuscipt, and resubmited a thoroughly improved version.

Author Response

Point.1, line 8-22 in the abstract part are the background, and should be shorted to two to three lines, show more contents on your main results and new findings. In the abstract, the two R packages are not developed by this work, and should not be mentioned in abstracts. Please show how MARS-ANN and MARS-SVM have better performance.

Response 1: The abstract has been summarised.

R packages are developed by this work. Links of published R packages are given below

MARSANNhybrid: https://cran.r-project.org/package=MARSANNhybrid

MARSSVRhybrid: https://cran.r-project.org/package=MARSSVRhybrid

MARS-ANN and MARS-SVM have better performance compared to singular ANN and SVR models.

Point 2. In the introduction, in line 33, the word “share” seems improper; pay attention to the first letter of “Morphological” and “it” in line 36, such grammatical errors are prevalent in the manuscript. Line 30-46 is the first paragraph (para), the author tell us too many things, but not focal, I would suggest the author divided it into to part, with the first describing the importance of lentil, and the second about the linear models along with their shortcomings. Line 47-82 is the second para, the author raised many examples, but didn’t introduce whether the methods used in this study have been used before, and what should followers do to make improvements. Therefore, I can’t see the necessity and meaning of this work. Moreover, the author should introduce these models in this para. Line 83-91 should tell the objective of this study, especially their new ideas or thinkings. Bear in mind that the full names of each abbreviation should be provided when it first appears.

Response 2. All grammatical errors have been rechecked and corrections have been made in the revised manuscript.

As per the suggestion, lines 30–46 are divided into two parts in the revised manuscript.

Lines 47–82 have been rewritten. Shortcomings of ML models and the importance of variables in the section are added in the section.

The full names of each abbreviation are checked and revised according to their appearance in the revised manuscript.

Point 3. in line 94-96, if this data have been used in other publications, it is better to cite the references;  in line 98, if the author use all the 21 descriptors, the author should provide these descriptors, and how they were measured as supplementary (supp). Some part (Line 108-120, 126) of “2.2. Data processing, statistical analysis and input variable selection” should be moved to the part of ”results”, if these results were not provided in other findings (if yes, these results should be removed or briefly introduced). Some part (Line 120-125) should be moved to “discussion”. Some part (Line 134-145) should be moved to “introduction”.  Some part (Line 153-155) should be moved to the first part of “Materials of Methods”.

Response 3. Citation related to the dataset has been included in the revised manuscript.

A supplementary sheet regarding the descriptors has been prepared and will be submitted with the revised manuscript.

All the sections have been restructured in the revised manuscript.

Point 4.  What is the relationship between “2.3.1. Multiple linear Regression (MLR) model” and this work, many manuscripts can copy and use lines 158-162.  Lines 164-170 should be move to “Introduction”.  The descriptions of “is number of hidden nodes, is the number of hidden nodes” in lines 172-173 indicate that this is a careless manuscript.

Response 4. The study focused on ML models. For ease of understanding, the ML model, Multiple Linear Regression (MLR) model and related results have been removed from the manuscript.

172-173 have been corrected. "n is number of hidden nodes, m is the number of input nodes"

Reviewer 2 Report

Overview and general recommendation:

The authors investigated the use of ML models for predicting yield crop in a lentil crop.

While I see the interest and utility of using such techniques and the novelty of the paper, the required technical details on the way those models have been built are totally missing from the manuscript thus making it impossible to evaluate if those models offer replicable unbiased results.

Moreover, the English level of the manuscript needs to be significantly improved in several parts as it is below standard.

Comments:

1.     References are not presented in order

2.     There is an almost infinite amount of English mistakes throughout the manuscript, which therefore requires a deep revision

3.     Table 1 should present also the measurement units for each variable. Were the data distribution gaussian?

4.     Variables are normalized, but was this done before or after data splitting? If done before, then this is a serious case of data leakage

5.     Are results shown in Table 1 and Figure 1 related to the whole dataset or just the training set?

6.     Are the results shown in the Results section obtained on the test set or on the whole dataset?

7.     Significant information is missing on the way the models have been built; without this info it is not possible to understand if the authors built models affected by overfitting

Author Response

Point 1. References are not presented in order

Response 1: All references are thoroughly reviewed and presented in a proper order in the revised manuscript.

Point 2. There is an almost infinite amount of English mistakes throughout the manuscript, which therefore requires a deep revision

Response 2: The manuscript has been thoroughly revised and all necessary corrections have been made.

Point 3. Table 1 should present also the measurement units for each variable. Were the data distribution gaussian?

Response 3: The measurement units for each variable are incorporated in the revised manuscript.

The data do not follow Gaussian distribution. Machine learning models do not require any prior assumption data distribution. Therefore, we use these models in the present study.

Point 4. Variables are normalized, but was this done before or after data splitting? If done before, then this is a serious case of data leakage.

Response 4: Yes. Variables were normalized after data splitting. The data were analysed using minimax normalization. To make the final prediction, all predicted values were denormalized after the prediction.

Point 5. Are results shown in Table 1 and Figure 1 related to the whole dataset or just the training set?

Response 4: Table 1 (Revised Table 2) and Figure 1 (Revised Figure 2) are related to the entire dataset. Table 1 represents summary of the data set, where the correlations of different variables with the study variable (yield) was visualized in figure 1.

Point 6. Are the results shown in the Results section obtained on the test set or on the whole dataset?

Response 6: The model building for the results section was done using train data, and all of the study results were based on test data.

Point 7. Significant information is missing on the way the models have been built; without this info it is not possible to understand if the authors built models affected by overfitting.

Response 7. Figure 1 depicts the process of the study. The two primary issues with the study were first, the use of MARS to identify key variables, and second, the use of these variables as input for ML models like ANN and SVR. The suitable MARS model for variable selection should be chosen before making any selections. We must choose the best MARS model for variable selection prior to making our choice of variables. The whole data set was spilt into training and testing sets. On the basis of training data, several orders of MARS models were developed, and test data was used to evaluate them. The MARS model that results in the lowest values for the performance metrics specified is considered as the variable selection model. In the entire set of data, this model was used to extract significant variables from the independent variables.

The chosen variables were then employed as inputs in ML models for yield prediction after variable selection. Data was normalised and split into training and testing sets for ML models. On the basis of the training data, the models were created, and the testing data was used to assess their generalizability.

Using a trial-and-error approach, we looked for the ideal set of ML model hyperparameters. In order to prevent overfitting, three folds (k = 3) were examined in this study for each dataset.

Reviewer 3 Report

Could you please explain why you chose these parameter for hyperparametrization process?

Author Response

Point 1. Could you please explain why you chose these parameter for hyperparametrization process?

Response 1: The hyperparametrization process is an important procedure as it helps to improve the performance of a machine learning model. In addition, it can help reduce overfitting and make the model more robust to changes in the data. Using a trial-and-error approach, we looked for the ideal set of ML model hyperparameters. In order to prevent overfitting, three folds (k = 3) were examined in this study for each dataset.

The optimal values of the hyperparameters are generally selected by minimizing a function of the error, in this study we have assumed the function to be “mean square error”. The values of the hyperparameters so obtained for ANN and SVR models are reported in Table 2.

Round 2

Reviewer 1 Report

I would suggest the author move table 7 and 8 to the part of results, and add new discussions.

Author Response

Response to Reviewer 1 Comments

Point 1: I would suggest the author move table 7 and 8 to the part of results, and add new discussions.

Response 1: The table 7 and 8  have been added to results section of the manuscript and discussions have modified.

Reviewer 2 Report

I am fine with all the modifications carried out by the authors

Author Response

Response to Reviewer 2 Comments

Point 1: I am fine with all the modifications carried out by the authors.

Response 1: Thank you for agreeing to our modifications.